# *Scutellaria barbata* D. Don Inhibits the Main Proteases (M^pro^ and TMPRSS2) of Severe Acute Respiratory Syndrome Coronavirus 2 (SARS-CoV-2) Infection

**DOI:** 10.3390/v13050826

**Published:** 2021-05-02

**Authors:** Sheng-Teng Huang, Yeh Chen, Wei-Chao Chang, Hsiao-Fan Chen, Hsiang-Chun Lai, Yu-Chun Lin, Wei-Jan Wang, Yu-Chuan Wang, Chia-Shin Yang, Shao-Chun Wang, Mien-Chie Hung

**Affiliations:** 1School of Chinese Medicine, China Medical University, Taichung 40402, Taiwan; sheng.teng@yahoo.com; 2Department of Chinese Medicine, China Medical University Hospital, Taichung 40402, Taiwan; u9702451@cmu.edu.tw (H.-C.L.); yclinjoyce@gmail.com (Y.-C.L.); 3Research Cancer Center for Traditional Chinese Medicine, Department of Medical Research, China Medical University Hospital, Taichung 40402, Taiwan; 4An-Nan Hospital, China Medical University, Tainan 709, Taiwan; 5Research Center for Cancer Biology, China Medical University, Taichung 40402, Taiwan; bluecrystalprotein@gmail.com (Y.C.); clamp0525@gmail.com (H.-F.C.); scpwang@gmail.com (S.-C.W.); 6New Drug Development Center, China Medical University, Taichung 40402, Taiwan; 7Graduate Institute of New Drug Development, China Medical University, Taichung 40402, Taiwan; ycwang322@gmail.com (Y.-C.W.); xin740103@hotmail.com (C.-S.Y.); 8Center for Molecular Medicine, China Medical University Hospital, Taichung 40402, Taiwan; proma1007@gmail.com; 9Department of Biological Science and Technology, China Medical University, Taichung 40402, Taiwan; cvcsky@gmail.com; 10Graduate Institute of Biomedical Sciences, China Medical University Taichung 40402, Taiwan; 11Department of Biotechnology, Asia University, Taichung 41354, Taiwan

**Keywords:** severe acute respiratory syndrome coronavirus 2 (SARS-CoV-2), pandemic, *Scutellaria barbata*, viral pseudoparticles

## Abstract

In late 2019, the severe acute respiratory syndrome coronavirus 2 (SARS-CoV-2) pandemic emerged to severely impact the global population, creating an unprecedented need for effective treatments. This study aims to investigate the potential of *Scutellaria barbata* D. Don (SB) as a treatment for SARS-CoV-2 infection through the inhibition of the proteases playing important functions in the infection by SARS-CoV-2. FRET assay was applied to investigate the inhibitory effects of SB on the two proteases involved in SARS-CoV-2 infection, M^pro^ and TMPRSS2. Additionally, to measure the potential effectiveness of SB treatment on infection inhibition, cellular models based on the Calu3 and VeroE6 cells and their TMPRSS2- expressing derivatives were assessed by viral pseudoparticles (Vpp) infection assays. The experimental approaches were conjugated with LC/MS analyses of the aqueous extracts of SB to identify the major constituent compounds, followed by a literature review to determine the potential active components of the inhibitory effects on protease activities. Our results showed that SB extracts inhibited the enzyme activities of M^pro^ and TMPRSS2. Furthermore, SB extracts effectively inhibited SARS-CoV-2 Vpp infection through a TMPRSS2-dependent mechanism. The aqueous extract analysis identified six major constituent compounds present in SB. Some of them have been known associated with inhibitory activities of TMPRSS2 or M^pro^. Thus, SB may effectively prevent SARS-CoV-2 infection and replication through inhibiting M^pro^ and TMPRSS2 protease activities.

## 1. Introduction

Emerging in late 2019, the spread of severe acute respiratory syndrome coronavirus 2 (SARS-CoV-2), named coronavirus disease-2019 (COVID-19) by the World Health Organization (WHO), rapidly reached global pandemic proportions. By late 2020, the total number of confirmed COVID-19 cases worldwide had reached over 72 million, with total reported deaths of more than 1.6 million [1]. The pandemic outbreak has severely impacted governments and health care systems around the world [2,3]. However, there currently exist only supportive therapeutic strategies to manage COVID-19 infection [4], while targeted antiviral drugs have yet to be developed [5]. As an ancient therapeutic strategy, traditional Chinese medicine (TCM) has historically played an important role in the treatment and prevention of diseases [6,7].

*Scutellaria barbata* D. Don (SB), also known as “Ban Zhi Lian”, is a widely used herb in TCM, demonstrating heat-clearing and detoxifying properties [8]. Neo-clerodane diterpenoids and flavonoids are the main compounds of SB, followed by polysaccharides, volatile oils, and steroids. In recent decades, studies of SB have focused on its anticancer, antioxidant, anti-inflammatory, antiviral, and antibacterial activities [8,9]. Most research into the cytotoxic effects (anti-tumor effects) of neo-clerodane diterpenoids, both in vitro and in vivo, have been related to hepatic cancer, breast cancer, colorectal cancer, lung cancer, and ovarian cancer [10,11,12,13,14]. Guo et al. reported that the total flavone could prevent anti-parainfluenza viral infection, and improve the membrane fluidity of Hep-2 cells [15]. Meanwhile, Wu et al. identified 11 diterpenoids that exhibited moderate to potent inhibition activities against Epstein–Barr virus lytic replication [16]. Antibacterial effects have also been reported, wherein SB showed an inhibitory effect against the extensively drug-resistant *Acinetobacter baumannii* [17]; in addition, gram-positive bacteria have demonstrated sensitivity to inhibition by SB [18]. Furthermore, apigenin and luteolin isolated from SB were found to be selectively toxic to the methicillin-resistant S. aureus and methicillin-sensitive S. aureus strains [19]. Other conditions, including diabetic retinopathy [20], cognitive dysfunction [21,22], and myocardial ischemia–reperfusion injury [23] have also been shown to benefit from SB treatment.

From the beginning of the SARS-CoV-2 pandemic, scientists have been eagerly investigating the key infection routes and viral infection cycle, with the aim of developing effective targeted antiviral treatments. SARS-CoV-2 uses the angiotensin converting enzyme II (ACE2) receptor to enter the host cell, in synergy with transmembrane serine protease 2 (TMPRSS2) [24]. TMPRSS2 is a cell-surface protein located on respiratory and gastroenterology epithelial cells; importantly, the coronavirus, including SARS-CoV-2, are known to enter human cells through interaction of viral spike protein with host ACE2 then cleaved by host TMPRSS2 protease. Thus, the rapid spread of communal infection is reliant on the efficiency of the ACE2-spike interaction and TMPRSS2 protease activity [25]. Meanwhile, the TMPRSS2: ERG fusion gene has been associated with prostate cancer and serves as a biomarker for the diagnosis and stratification of androgen-sensitive prostate cancer [26]. Although earlier studies primarily focused on prostate cancer, more recent studies have investigated TMPRSS2/ACE2 integrity as related to SARS-CoV-2 infection. Several studies have indicated that modifications to the renin-angiotensin pathway or TMPRSS2 expression would break down viral infections [25,27,28].

The screening of natural compounds in several studies, including 10-hydroxyusambarensine and cryptospirolepine, has revealed the potential of inhibiting host cell entry by altering the ACE2-TMPRSS2 pathway [29,30,31]. Compounds inhibiting TMPRSS2 protease activity have notably been associated with reduced infection activity of the SARS-CoV-2 Vpp [32]. Additionally, more recent investigations have been conducted to determine effects related to M^pro^, also called 3C-like protease (3CL^pro^), one of the SARS-CoV-2 nonstructural proteins. M^pro^ cleaved the original viral polyproteins—pp1a and pp1ab—with its proteolytic activity. Of note, the 33.8-kDa M^pro^ is necessary for viral replication and transcription [33]. Recent research has focused on developing a screening strategy for effective M^pro^ inhibitors, potentially associated with SARS-CoV-2 infection inhibition [34,35]. In structural based screenings, natural compounds including epigallocatechin gallate, epicatechin gallate, gallocatechin-3-gallate, glycyrrhizin, bicyclogermacrene, tryptanthrine, β-sitosterol, indirubin, indican, indigo, hesperetin, chrysophanic acid, rhein, berberine, and β-caryophyllene have been identified as potential herbal M^pro^ inhibitors against SARS-CoV-2 infection [36,37]. Furthermore, herbs including *Glycyrrhiza, Tinospora cordifolia, Uncaria tomentosa* have demonstrated M^pro^ inhibitory effects [38,39,40].

Herbal medicines are often considered as low-toxicity alternative treatment modalities, while various herbal compounds are regarded as potential treatments for SARS-CoV-2. Recent studies have revealed the effectiveness of several natural compounds at TMPRSS2 or M^pro^ inhibition [30,32,38,41]. However, few studies investigating TCM herbs have been reported. Indeed, previous studies have shown that SB (Ban-zhi-lian) exhibits notable antiviral, antibacterial, anticancer, and immunomodulation effects; although investigation into the potential anti-SARS-CoV-2 effects of SB and its role in TMPRSS2 and M^pro^ inhibition remains limited. Thus, this study aims to evaluate the effectiveness of SB on the inhibition of M^pro^ and TMPRSS2 associated with SARS-CoV-2 infection activity. In this current study, we show that SB inhibits protease activity of SARS-CoV-2 M^pro^ and TMPRSS2 and suppresses pseudoviral entry, paving a way for therapeutic application as an anti-SARS-CoV-2 agent.

## 2. Materials and Methods

### 2.1. Preparation of *Scutellaria barbata* (SB) Aqueous Extract

The SB utilized in this study is recognized in accordance with the definition described in Flora of Taiwan, with a sample of SB deposited in Set D, TN: BRCM 4616 in the herbarium of National Taiwan University. The extract was prepared from the whole plant according to Taiwanese GMP methods and guidelines, and the concentrated powder was manufactured by a recognized GMP company (Kaiser Pharmaceutical Co., Ltd., Tainan, Taiwan). Of note, it is applied in clinical practice and covered by the National Health Insurance system of Taiwan. Briefly, the 2.5 g powder was boiled with 10 mL ddH_2_O for 10 min, followed by centrifugation of 3000 rpm for 10 min and 12,000 rpm for 15 min. The resulting extract was filtered with 0.22 μm microfilter and then aliquot for reservation. The extraction rate was 14.2%, i.e., 1 g of concentrated powder (extract: excipient 1: 1) is equal to 3.5 g of the original herb. The stock concentration was 400 mg/mL.

### 2.2. Liquid Chromatography–Mass Spectrometry (LC/MS) Analysis

The aqueous extracts of SB were analyzed using a Velos Pro dual-pressure linear ion trap mass spectrometer (Thermo Fisher Scientific, San Jose, CA, USA) equipped with an Agilent 1100 Series binary high-performance liquid chromatography pump (Agilent Technologies, Palo Alto, CA, USA), and a Famos autosampler (LC Packings, San Francisco, CA, USA), using a Xbridge column (1 mm I.D. × 150 mm, 3.5 um, 130 Å). Briefly, the gradient employed was 2% buffer B at 2 min, to 98% buffer B at 20 min, with a flow rate of 50 μL/min, where buffer A was 0.1% formic acid/H_2_0 and buffer B was 0.1% formic acid/acetonitrile. The survey scan was acquired in the mass range *m*/*z* 200–2000. Electrospray voltage was maintained at 4 kV and capillary temperature was set at 275 °C.

### 2.3. Estimation of the Average Molecular Weight of Aqueous Extract of SB

The mass range 200–2000 was divided into 45,000 intervals and the mass intensities in each interval were summed from 5340 scan events during a 30 min analytical process. The relative intensity ratio of each interval was acquired by the calculation of (an induvial mass intensity/total mass intensity) as shown in Appendix A.

$\sum_{i=1}^{5340}\left[{\left(\mathrm{mass}\right)}_{i}\times {\left(\mathrm{intensity}\;\mathrm{ratio}\right)}_{i}\right]$. The average molecular weight = 547.6 Da.

### 2.4. Protein Sample Preparation

Recombinant SARS-CoV-2 M^pro^ and its protein substrate were prepared following the previously described protocol [41]. Briefly, the target gene encoding SARS-CoV-2 M^pro^ (residues 3264–3569, UniProt accession: P0DTD1) and the fluorescent protein substrate of M^pro^ (CFP-TSAVLQSGFRKM-YFP) were synthesized and subcloned into pSol SUMO vector (Lucigen) and pET16b vector, respectively. Overexpression of SARS-CoV-2 M^pro^ was induced by 0.5 mM IPTG until O.D._600_ reached 0.6 and further incubated for 18 h at 20 °C, while its fluorescent protein substrate was induced by 20% L-rhamnose and incubated for 20 h at 16 °C. Human TMPRSS2 (amino acids 256-492, UniProt accession: C9JKZ3) with N-terminal His_6_-MBP-tag and its protein substrate (GFP- QTQTNSPRRARSVAS-RFP) were constructed and recombinantly expressed in *E. coli* BL21 (DE3) as previously described [41]. All the protein samples were purified by immobilized-metal affinity chromatography using ÄKTA™ go chromatography system (Cytiva) according to manufacturer instructions. The purity and monodispersity of protein samples were further achieved by an additional purification step using size-exclusion chromatography (HiLoad Superdex 200, GE Healthcare, Chicago, IL, USA).

### 2.5. Fluorescence Resonance Energy Transfer (FRET) Assay

The enzyme activity of SARS-CoV-2 M^pro^ was determined by FRET assay, as previously detailed [41]. For the activity assay of TMPRSS2, 100 μL reaction volume containing 25 μM TMPRSS2 protein in the presence or absence of SB was pre-incubated at room temperature for 30 min. The reaction was initiated by addition of the fluorescent protein substrate and monitored continuously for 2 h at 536 nm after excitation at 506 nm using a Synergy H1 microplate reader (BioTek Instruments, Inc., Winooski, VT, USA). The concentrations of SB used in the FRET assays of SARS-CoV-2 M^pro^ were from 0.5 mg/mL to 4 mg/mL with two-fold series dilution. The IC_50_ value of SB on inhibition of SARS-CoV-2 M^pro^ was determined by using 7 different concentrations of SB and calculated by Prism 9 software. The final concentration of SB used in the FRET assays of TMPRSS2 was 4 mg/mL.

### 2.6. Cells

Calu3 cells were cultured in Modified Eagle’s Medium (MEM) supplemented with 10% fetal bovine serum (FBS), 1 × NEAA, and 1% penicillin/streptomycin. VeroE6 were cultured in Dulbecco’s Modified Eagle’s Medium (DMEM) supplemented with 10% FBS, 1× GlutaMAX, and 1% penicillin/streptomycin. TMPRSS2 expressing cells of Calu3 and VeroE6 cell lines were generated by transfection of the pCMV3-TMPRSS2-Flag plasmid and selection by hygromycin.

### 2.7. Viral Pseudoparticles (Vpp) Infection and Inhibition

For the infection experiments, 10,000 cells were seeded in 96-well plates. The cells were then pre-incubated with a medium containing different concentrations of SB or H_2_O (vehicle control) for 1 h at 37 °C and 5% CO^2^, before they were infected with the Vpp harboring SARS-CoV-2-S and luciferase reporter (purchased from National RNAi Core Facility (NRC), Academia Sinica, Taipei, Taiwan), followed by centrifugation at 1250 *g* for 30 min. After 24 h incubation, the Cell Counting Kit-8 (CCK-8) assay (Dojindo Laboratories, Mashiki, Kumamoto, Japan) was performed to measure cell viability. Then, each sample was mixed with an equal volume of ready-to-use luciferase substrate Bright-Glo Luciferase Assay System (Promega, Madison, WI, USA), with luminescence measured immediately by the GloMax Navigator System (Promega, Madison, WI, USA). Raw luminescence values, indicating luciferase activity, were recorded as counts per second. The relative light unit (RLU) was normalized with cell viability first, then the control group was set as 100%, and the relative infection efficiencies were calculated.

### 2.8. Statistical Analysis

All statistical analyses were calculated with SigmaStat statistical software (version 2.0, Jandel Scientific, SanRafael, CA, USA). All data originating from experiments were performed in triplicate and represented as mean ± standard deviation (SD). Statistical significance was calculated using student *t*-test. ANOVA was performed when multiple comparisons were evaluated. *P* values less than 0.05 were considered to be significant. All experiments were repeated at least three times independently.

## 3. Results

### 3.1. Initial Screening of Scutellaria barbata against SARS-CoV-2 M^pro^ and TMPRSS2

For the initial screening, we selected twenty herbs in two independent batches to identify protease enzyme activities against SARS-CoV-2 M^pro^. One of these herbs, *Scutellaria barbata* (Ban-zhi-lian), which is utilized in treating liver, lung, and colon cancer [42,43,44], was determined the best candidate for SARS-CoV-2 inhibition. We, therefore, investigated the aqueous extracts of SB on the inhibition of the main protease activity of SARS-CoV-2, a primary therapeutic target due to its critical role in viral replication. The FRET assay revealed that 4 mg/mL SB effectively inhibited the SARS-CoV-2 M^pro^, with a remarkable inhibition rate of 93.3% (Figure 1). To further evaluate the efficacy of the aqueous extracts of SB, we performed a dose–response titration enzymatic assay. The results showed that 2 mg/mL, 1 mg/mL, and 0.5 mg/mL SB inhibited the protease activity of SARS-CoV-2 M^pro^ at rates of 79.5%, 30.1%, and 1.6%, respectively (Figure 1). The half-maximal inhibitory concentration (IC_50_) of SB on inhibition of SARS-CoV-2 M^pro^ was further determined to be 1.27 mg/mL (calculated by using the average molecular weight of 547.6 Da of SB) (Figure 2). Another important therapeutic target to combat SARS-CoV-2 is TMPRSS2, which plays an essential role in the cellular entry of SARS-CoV-2. The in vitro enzymatic assay demonstrated that 4 mg/mL SB had a moderate inhibitory effect (54.8%) on TMPRSS2 activity (Figure 3).

### 3.2. Scutellaria barbata Inhibited Pseudovirus Infection through TMPRSS2

SARS-CoV-2 can enter the host cell via both endosomal and non-endosomal pathways. The endosomal-mediated pathway includes the S protein binding to ACE2 and subsequent dynamin/clathrin-mediated internalization of endosomal vesicles. In addition to the endosomal-mediated pathway, host TMPRSS2 also play critical roles in the non-endosomal entry of SARS-CoV-2 [1]. VeroE6 cell line (monkey kidney epithelial cell line) is used as a standard model for endosomal entry pathway, because this cell line presents relatively high ACE2 expression and low TMPRSS2 expression. Calu3 cell line (human lung cancer cell line), as with the airway epithelial cells, express high amounts of TMPRSS2 and SARS-CoV-2 entry into these cells is dependent on TMPRSS2 [45]. Calu3 and VeroE6 cells with and without TMPRSS2 over-expression were treated with *S**cutellaria barbata*. After 24 h treatment, the cell viability measured by CCK8 assay demonstrated no cytotoxicity for both (Appendix A). Our results already demonstrated that SB could potently inhibit TMPRSS2 enzyme activity in vitro (Figure 3). To further investigate whether SB would block SARS-CoV-2 infection through inhibiting TMPRSS2 activity, we first generated the TMPRSS2-expressing cells in both VeroE6 and Calu3 cells. Then, these cells were pre-treated for 1 h with 4 or 8 mg/mL of SB and then infected with the SARS-CoV-2 Vpp and infection efficiency was determined according to luciferase activities after 24 h of infection. The results demonstrated that SB blocked infection of the SARS-CoV-2 Vpp in Calu3 cells, but not in VeroE6 cells (Figure 4). Furthermore, SB showed a higher inhibitory effect in both over-expressing TMPRSS2 cells (Figure 4). These results indicate that SB can inhibit SARS-CoV-2 entry by affecting TMPRSS2 activity.

### 3.3. The Scutellaria barbata Fingerprint

To analyze the composition of SB, the aqueous extracts of SB were separated by HPLC (high performance liquid chromatography), and the exact molecular weights of the individual constituent compounds were determined by mass spectrometry. The major ingredients identified, including apigenin, naringenin, scutellarin, baicalein, luteolin, and wogonin are consistent with previous reports [8,46], as shown in Figure 5. It is interesting to note that several derivatives of SB, including scutellarin, baicalein, wogonin, and luteolin have also been reported to inhibit M^pro^ and TMPRSS2 (please see discussion), which provides a scientific basis for the SB to inhibit SARS-CoV-2 infection.

## 4. Discussion

The main protease (M^pro^) functions in a dimeric form, involving in proteolytic processing of replicase polyproteins in SARS-CoV-2. Drugs targeting M^pro^ with broad-spectrum antiviral activity and lower the risk of mutation-mediated drug resistance are potent antiviral agents against SARS-CoV-2 infection [47]. Based on M^pro^ inhibitors of TCM herbs screening, *Zingiberis Rhizoma Recens, Asteris Radix et Rhizoma, Notoginseng Radix et Rhizoma, Chuanxiong Rhizoma, Salviae Miltiorrhizae Radix et Rhizoma, Zingiberis Rhizoma, Dianthi Herba, Rhei Radix et Rhizoma, Cistanches Herba* are reported by Ma et al. [48]. In this study, we found that SB effectively inhibited the SARS-CoV-2 M^pro^. Additionally, Xia et al. found that Shufeng Jiedu capsules can reduce the activity of NFκB in the lungs of HCoV-229E mice. Its active components, wogonin, quercetin, and polydatin, bind directly to the SARS-CoV-2 M^pro^ [49].

Spike-ACE2 interaction and TMPRSS2 protease activity are critical for host cell-mediated viral entry of SARS-CoV-2. Thus, drugs which inhibit TMPRSS2 or spike-ACE2 interaction, thereby blocking viral fusion, are of particular interest in the search for SARS-CoV-2 treatments. Various existing drugs, including chloroquine, hydroxychloroquine, and other antiviral drugs such as remdesivir (a nucleotide analogue for Ebola virus), lopinavir and ritonavir (HIV protease inhibitors), and arbidol and favipiravir (broad-spectrum anti-influenza drugs) were early potential treatment candidates for SARS-CoV-2 [50]. Meanwhile, bromhexine hydrochloride is a mucolytic cough suppressant, and as a TMPRSS2 inhibitor, may hold potential for treatment of SARS-CoV-2 infection [51]. Another drug, camostat, which is used to treat chronic pancreatitis, could inhibit SARS-CoV-2 infection via serine protease effects [52]. Notably, 10–hydroxyusambarensine, cryptospirolepine, and cryptoquindoline, which are extracted from African medicinal plants, are reported to exhibit superior TMPRSS2 inhibitory effects [31]. Of those three compounds, 10–hydroxyusambarensine was identified as the most effective, by docking into the S1-specificity pocket of TMPRSS2 [31]. Other natural compounds, including geranium, lemon essential oils, homoharringtonine, and halofuginone have also demonstrated TMPRSS2 inhibitory effects [30,53]. Of note, several derivatives of SB, including scutellarin, baicalein, wogonin, and luteolin have also been reported to inhibit M^pro^ and TMPRSS2, further supporting the promising results reported herein. Moreover, a recent study indicated another host protease, furin, is also involved in virus infection via a separate mechanism [54], although it is currently unclear whether SB may also exert an effect on a furin-mediated pathway.

This is the first study to report on the role of SB against SARS-CoV-2 infection through the inhibition of the main protease. The FRET assay demonstrated that SB blocked SARS-CoV-2 infection through inhibition of M^pro^ and TMPRSS2. Furthermore, in Calu3 cells and over-expressing TMPRSS2 cells, SB treatments inhibited the SARS-CoV-2 Vpp infection. These results indicate the potential of SB as an alternative drug to treat SARS-CoV-2 infection. Several compounds present in SB have recently been recognized for their potential benefits in clinical application. As a major compound of SB, scutellarin has exhibited NLRP3 inflammasome suppression, thereby rescuing mice from bacterial sepsis [55]. In addition, a silico screening report of scutellarin revealed M^pro^ inhibitory effects, supporting our study results [56]. Meanwhile, baicalein, which is a flavone extracted from the roots of *Scutellaria baicalensis* and *Scutellaria lateriflora*, has exhibited notable anti-oxidant, anti-virus, anti-bacteria, anti-inflammatory, anti-cancer, and immunoregulatory effects [57,58,59]. Huang et al. have reported that natural compounds including baicalein, luteolin, naringenin, and wogonin suppress ACE2 and M^pro^ protease activity to suppress SARS-CoV-2 [60]. Baicalein, a main ingredient of the TCM herbal formula Shuanghuanglian, has demonstrated anti-viral effects through inhibiting M^pro^ activity [61,62]. Additionally, studies have reported that baicalein significantly affects both ACE2 and TMPRSS2 [63,64]. A notable screening study, suggesting the compound as a candidate for the treatment of SARS-CoV-2, reported that baicalein also binds with the SARS-CoV-2 nonstructural protein 14 (NSP14) [65]. Another constituent compound of SB is luteolin, a yellow dye compound that is obtained primarily from *Reseda luteola*, is in fact an ingredient in many foods, including celery, broccoli, parsley, chamomile tea, carrots, olive oil, peppermint, oranges, and medicinal herbs such as *Artemisia*. Luteolin has been shown to exhibit anti-inflammatory, anti-apoptosis, and anti-cancer effects [66,67,68]. Furthermore, luteolin, one of the major compounds in Lianhua Qingewen capsules and Maxingyigan, has been reported as a potential treatment candidate of SARS-CoV-2 via targeting of the Akt1 (serine/threonine kinase 1) pathway [69]. Liang et al. also demonstrated that luteolin could inhibit SARS-CoV-2 by interleukin-6 down-regulation [70]. Luteolin has been associated with anti-coronavirus effects via both inhibiting ACE2 and M^pro^ activity, further confirming our results [71,72,73]. Another constituent component isolated from SB, wogonin, is a key flavonoid for quality control and exhibits anti-inflammatory, anti-cancer, anti-viral, anti-angiogenesis, anti-oxidant, and neuro-protective effects [8]. Together with phytol, luteolin, and hispidulin, wogonin is one of the active constituents in the chloroform fraction of SB, which has demonstrated dose-dependent cytotoxicity on six cancer cell lines [74]. Wogonin presents anti-viral activities against influenza infection via modulation of AMP-activated protein kinase (AMPK) pathways, and anti-fibrotic effects in liver fibrosis via regulating the activation and apoptosis of hepatic stellate cells [75]. Thus, wogonin may present a therapeutic option to attenuate the cytokine storm in acute symptomatic SARS-CoV-2 infections due to the immunomodulatory response [76] and its anti-fibrotic effects in the convalescence period [75]. Yet another flavonoid compound for quality control in SB [8], naringenin, can inhibit two-pore channels (TPCs) [77,78], which regulate the trafficking of the virus to late-endosomes/lysosomes following entry into cells [79]. Several studies have investigated and reported on the antiviral activities of naringenin against hepatitis C (HCV), influenza A, Zika, and Dengue Fever [80,81,82,83]. Thus, it is reasonable to suggest that naringenin may also be effective in combating SARS-CoV-2. Finally, the compound apigenin has been shown to suppress influenza A virus-induced RIG-I activation and viral replication [84], and EBV reactivation [85]. It also exhibits an inhibitory effect on the foot-and-mouth disease virus [86], the African swine fever virus [87], the buffalopox virus [88], and the enterovirus [73,89]. However, it must be noted that in order to reach effective plasma concentrations. There is a potential limitation to take a relatively high dose of SB in vivo. Nevertheless, taken together, SB and its major constituent components indeed demonstrate various anti-viral effects which thus warrant consideration for inclusion in the development of future therapeutic strategies for SARS-CoV-2 infection.

## 5. Conclusions

SARS-CoV-2 is an RNA-virus currently challenging health care systems worldwide. As candidates for SARS-CoV-2 prevention and treatment, anti-viral herbal compounds present low-cost and low-toxicity options. The anti-viral, anti-bacterial, anti-inflammation, and anti-cancer effects of SB indicate the therapeutic promise of SB. This is the first study to report that SB effectively inhibits M^pro^ and TMPRSS2 activity in vitro. Inhibition of TMPRSS2 is able to suppress SARS-CoV-2 virus particle, Vpp to infect host cells. And suppression of M^pro^ activity is known to inhibit virus replication. Thus, SB may associate with dual activities to inhibit both SARS-CoV-2 virus infection and replication. Our results demonstrate that SB would be an ideal candidate for inclusion in the development of SARS-CoV-2 treatments. Further investigation into the clinical efficacy of SB associated with its distinct molecular mechanisms is thus warranted.

## Figures and Tables

**Figure 1 viruses-13-00826-f001:**
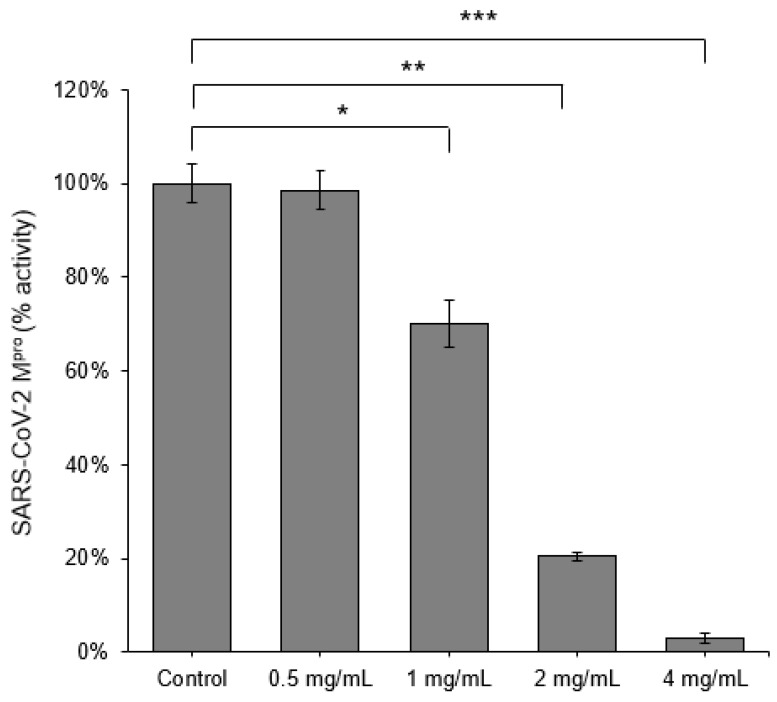
*Scutellaria barbata* inhibited the protease activity of SARS-CoV-2 M^pro^ in vitro FRET-based enzyme activity assay of SARS-CoV-2 M^pro^ in the absence or presence of *Scutellaria barbata* treatment with two folds series dilution from 0.5 mg/mL to 4 mg/mL. All data were represented as mean ± standard deviation (SD). Statistical significance was calculated using student *t*-test. ANOVA was performed when multiple comparisons were evaluated. *, *p* < 0.05. **, *p* < 0.01. ***, *p* < 0.001.

**Figure 2 viruses-13-00826-f002:**
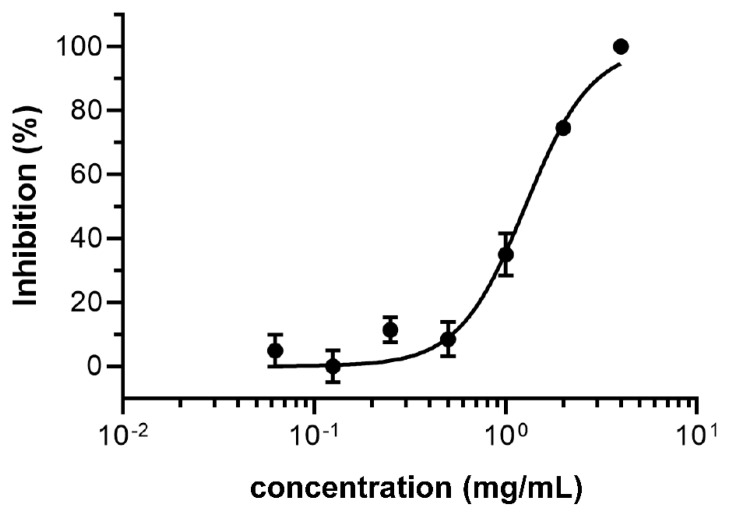
Dose-response curve of SB on SARS-CoV-2 M^pro^. SB inhibited SARS-CoV-2 M^pro^ with half-maximal inhibitory concentration (IC_50_) of 1.27 mg/mL and Hill slope of 2.5.

**Figure 3 viruses-13-00826-f003:**
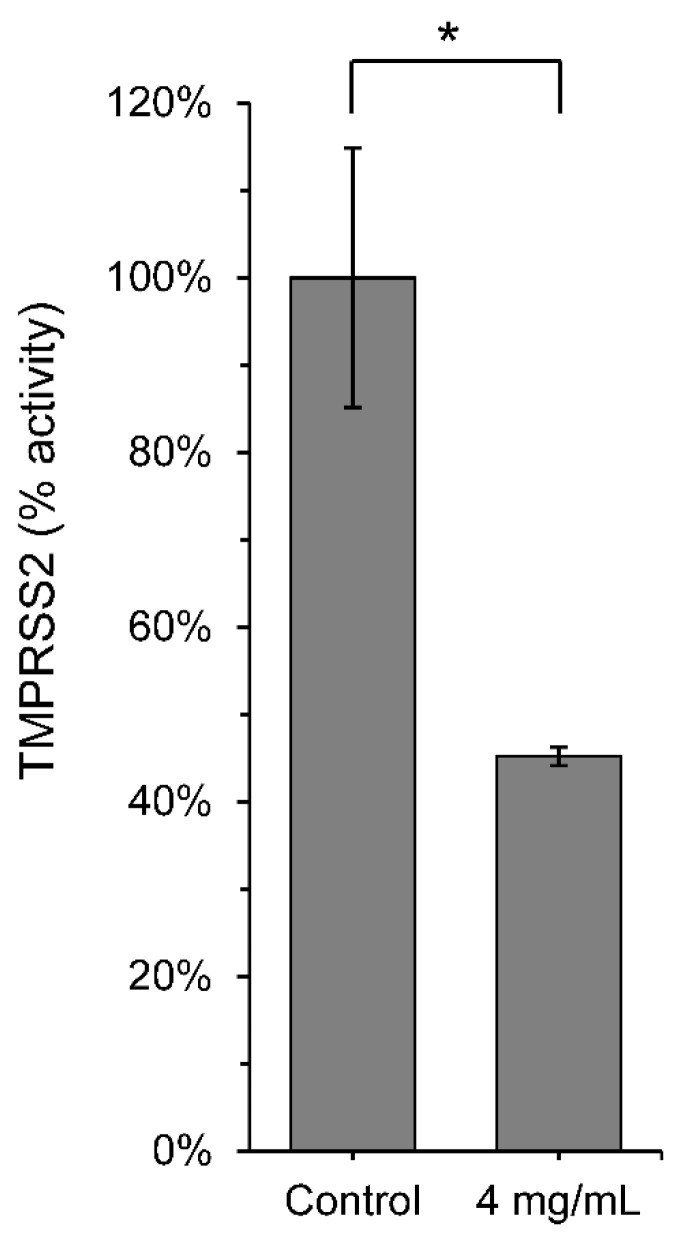
*Scutellaria barbata* inhibited the protease activity of TMPRSS2 in vitro FRET-based enzyme activity assay of TMPRSS2 in the absence or presence of 2 mg/mL *Scutellaria barbata* treatment. All data were represented as mean ± SD. Statistical significance was calculated using student *t*-test. * *p* < 0.05.

**Figure 4 viruses-13-00826-f004:**
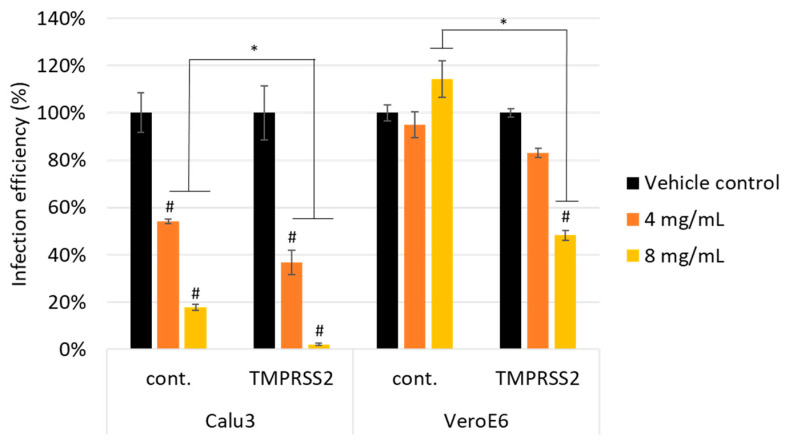
*Scutellaria barbata* inhibited pseudovirus infection through TMPRSS2. Calu3 and VeroE6 cells with and without TMPRSS2 over-expression were pre-treated with *Scutellaria barbata* and then infected with SARS-CoV-2 spike pseudovirus. After 24 h infection, infection efficiency rate was measured according to luciferase activities. All data were represented as mean ± SD. Statistical significance was calculated using student *t*-test. ANOVA was performed when multiple comparisons were evaluated. #, *p* value < 0.05 compared with vehicle control. *, *p* value < 0.05 indicates between control cells and TMPRSS2 over-expressing cells.

**Figure 5 viruses-13-00826-f005:**
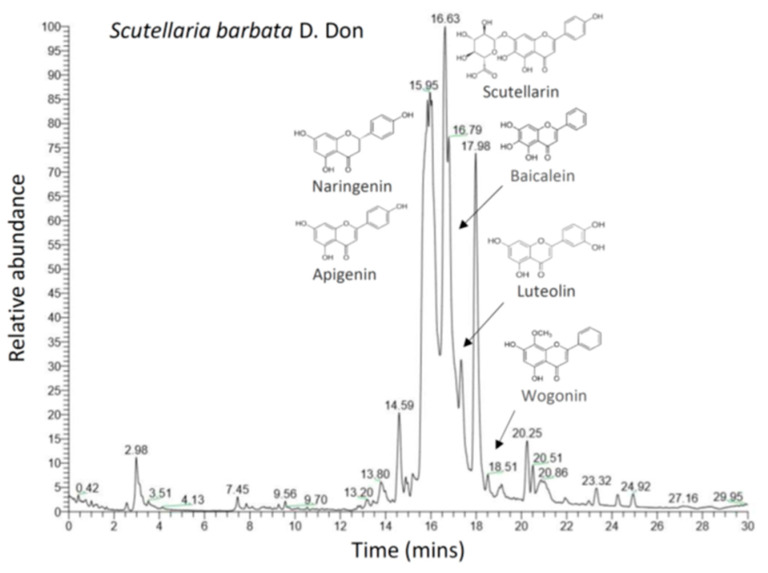
The fingerprint of *Scutellaria barbata.* Aqueous extracts of *Scutellaria barbata* was analyzed by LC/MS. The main compositions of individual herbal compounds were indicated by arrows based on the exact molecular weight.

## Data Availability

The data presented in this study are available in this article and on request from the author and corresponding author.

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
