# Peer review of "Scutellaria barbata D. Don Inhibits the Main Proteases (Mpro and TMPRSS2) of Severe Acute Respiratory Syndrome Coronavirus 2 (SARS-CoV-2) Infection"

_viruses, 2021, doi:10.3390/v13050826_

Round 1

Reviewer 1 Report

The manuscript by Sheng-Teng Huang et.al.  entitled ’Scutellaria barbata D. Don Inhibits the Main Proteases (Mpro and 2 TMPRSS2) of Severe Acute Respiratory Syndrome Coronavirus 2 (SARS-CoV-2) Infection’ focuses on the treatment of SARS-CoV-2 infection via naturally occurring  substances from herb. The manuscript is well written and consists of classical sections. The SARS-CoV-2 pandemic is scientific, medical and global social problem worldwide. Therefore, the finding of a new potential antiviral substance which is blocking SARS-CoV-2 entry to the cell can help to solve the global crisis caused by the virus outbreak.  The methods used in this work are properly chosen and performed according to the highest standards. The conclusions are supported by the presented results.

Minor comments:

  1. Some unfortunate typos should be corrected.

Author Response

Minor comments:

1. Some unfortunate typos should be corrected.

Reply: Thanks for your reminder. We have carefully read and corrected the typos.

Reviewer 2 Report

In Fig.1. the authors used different concentration (mg/ml) of Scutellaria barbata (SB) Aqueous Extract to inhibit the activity of SARS-CoV-2 M protein.

However, the maximum half inhibition concentration of Scutellaria barbata (SB) Aqueous Extract on SARS-CoV-2 M protein is mM. I suggest that the authors used mg/ml as the concentration unit.

Please examine the cytotoxic effect of Scutellaria barbata (SB) aqueous extract on cells

Author Response

In Fig.1. the authors used different concentration (mg/ml) of Scutellaria barbata (SB) Aqueous Extract to inhibit the activity of SARS-CoV-2 M protein. However, the maximum half inhibition concentration of Scutellaria barbata (SB) Aqueous Extract on SARS-CoV-2 M protein is mM. I suggest that the authors used mg/ml as the concentration unit.

Reply: Thanks for your suggestion. We have changed the concentration unit of maximum half inhibition into mg/ml in Fig. 2.

Please examine the cytotoxic effect of Scutellaria barbata (SB) aqueous extract on cells.

Reply: Thanks for your suggestion. Calu3 and VeroE6 cells with and without TMPRSS2 over-expression were treated with Scutellaria barbata. After 24 hours treatment, the cell viability measured by CCK8 assay demonstrated no cytotoxicity for both (Supplementary Figure 1).
